# OpenReview forum: "COIR: Chain-of-Intention Reasoning Elicits Defense in Multimodal Large Language Models"
_NeurIPS.cc/2025/Workshop/Reliable_ML — NeurIPS 2025 - Reliable ML Workshop_

### Official Review · Reviewer_G3WT · 2025-09-12
**There is a lack of clarity regarding this work’s contribution to the existing literature.**

**Rating:** 5
**Confidence:** 4

**Review:**

The introduction is not compelling, as it fails to establish a clear connection between prior jailbreaking approaches and the proposed intention-based approach (why intention-based among so many potential approaches?). It gives the impression that the authors developed the idea separately and then merely summarized related jailbreaking techniques in the introduction.

Additionally, it is unclear whether similar work has been conducted for purely text-based (non-multimodal) LLMs. If not, it would be helpful to explain why the study begins with multimodal large language models rather than first exploring the approach on standard LLMs.

---

### Official Review · Reviewer_jDm4 · 2025-09-18
**Review of "COIR: Chain-of-Intention Reasoning Elicits Defense in Multimodal Large Language Models"**

**Rating:** 7
**Confidence:** 4

**Review:**

This paper proposes "Chain of Intention Reasoning", a prompt-based approach to detect adversarial prompts and refuse unsafe answers. This approach tries to reason the underlying intent of the input prompt. The paper showed that their approach outperforms other top approaches, including ECSO, AdaShield, etc, on MM-SafetyBench and HADES benchmark sets.

Strength:
* Simple yet efficient pro-defence approach via prompting.
* Consistent improvements in different topics on the Benchmark than other approaches
* Claimed that it does not over-defend and maintains competitive performance in regular i.e., benign tasks.

Weakness:
* A prefix-based evaluation approach may be an unreliable approach. The semantic approach seems more appropriate.
* A significant decrease in prefix-based score (83.66 to 69.72) on the "Political Lobbying" category using a larger model compared to a smaller model, which isn't discussed.
* Reporting statistical significance would be more helpful, i.e., including standard deviation, etc.
* Minor case: Spelling inconsistency of Adashield and AdaSheild.
* Since safeguards are deployed in real-time, it will be better to discuss the latency of COIR, since it requires a reasoning step in the output.
* Getting mostly 100% scores for Table 2 is also a concern since it is not usual for any benchmark for any ML tasks. It will be better to clarify this case (i.e., why it is saturated) in the paper.

---

### Official Review · Reviewer_42cS · 2025-09-20
**Practical simple prompt modification with demonstrable gains**

**Rating:** 7
**Confidence:** 3

**Review:**

# *Summary*

This paper shows that by modifying the prompt to also assess the users intention, we can elicit LLMs against jailbreaking better, compared to prompts without it.Method is Prompting only. Two components inside one prompt: a Chain-of-Thought rationale and a Chain-of-Intention part that explicitly infers intent and rates harmfulness across text and image, then either answers or refuses.

# *Strengths.*

**Method:**

- The paper tests a simple modification to prompts across a variety of scenarios, and shows how robust a simple change can be. Much appreciated. ( simplicity here is a strength)
- Simple, single-pass defense with no extra inference steps; directly practical.
- Ablations identify the marginal value of rationale and intention components.

**Writing:**

- Introduction is detailed in sharing context to the problem.
- Research questions explored are clear.
- Related work coverage is adequate.
- Paper is well organized and easy to follow.

- Good usage of examples.

# *Weaknesses / Limitations.*

I understand DSR as the proportion of harmful queries where the model refuses. But I think the question of whether this prompt modification leads to over cautiousness is not clearly answered. An assessment over a blend of harmful and non-harmful queries, and looking at False \+ve rate and Recall needs to be reported as well to complete the paper.

# *Suggestions for Authors.*

An analysis of errors would make the paper stronger.It would be interesting to see if the reasons why a model refuses to answer are the correct reasons in itself, but perhaps that is for future work.